# QUANTITATIVE LLM JUDGES

## ABSTRACT

LLM-as-a-judge is a framework where a large language model (LLM) evaluates the output of another LLM. While LLMs excel at producing qualitative textual evaluations, they often struggle to predict human preferences and numeric scores. We propose *quantitative LLM judges*, which align evaluation scores of existing LLM judges to humans in a given domain using regression models. The models are trained to improve the score of the original judge using its rationale and score. We present four quantitative judges for different types of absolute and relative feedback, which showcases the *generality* and *versatility* of our framework. Our framework is more *computationally efficient* than supervised fine-tuning and can be more *statistically efficient* when human feedback is limited, which is expected in practice. We validate these claims empirically on four datasets using two base judges. Our experiments show that quantitative judges can improve the predictive power of existing judges through post-hoc modeling.

## 1 INTRODUCTION

Measuring the quality of generated natural language presents several challenges due to the diverse range of generation methods and evaluation criteria (Gehrmann et al., 2023). The *LLM-as-a-judge* paradigm has recently emerged as a compelling approach to these evaluation challenges. By leveraging the reasoning capabilities of *large language models (LLMs)*, this approach can provide more nuanced assessments that correlate better with human judgments across diverse tasks (Gu et al., 2025). LLM judges typically output both rationales and numeric scores, thus combining the comprehensiveness of human evaluation with the scalability of automated metrics.

Recent studies highlight issues with LLM judges such as low alignment with human scores, miscalibration, score compression, high variance, prompt sensitivity, and leniency bias (Thakur et al., 2025; Wei et al., 2025). To address these issues, various approaches have been proposed to fine-tune LLMs to improve score prediction on specific tasks (Chiang et al., 2025; Lukasik et al., 2025). However, fine-tuning is often impractical, due to requiring large datasets and having a high computational cost. As a result, many users continue to rely on off-the-shelf models like GPT-4 or Claude (Bavaresco et al., 2024; Huang et al., 2024).

The main issue with current LLM judges is that they conflate qualitative reasoning with quantitative assessment. For example, fine-tuned LLM judges like Prometheus (Kim et al., 2024b) are trained with the cross-entropy (CE) loss on gold-standard textual evaluations to generate both qualitative rationales and numeric scores. This is an inherent mismatch in objectives, as LLMs optimized for next-token prediction are tasked with producing accurate numeric scores, a fundamentally different statistical problem. While LLMs excel at producing structured textual evaluations, they are poorly suited for regressing human scores or preferences (Chiang et al., 2025).

This mismatch raises a natural question: *can qualitative rationales be decoupled from quantitative scores to achieve more accurate judgment?* This decoupling allows the LLM produce high-quality rationales while numeric scores are predicted by classic machine learning models that are robust and loved by practitioners. This perspective is supported by prior works in interpretability and probing, which show that when model representations encode information relevant for downstream tasks, simple linear decoders can recover this information effectively (Alain and Bengio, 2017; Hewitt and Manning, 2019; Hupkes et al., 2018; Belinkov, 2022).

Building on this insight, we present *quantitative judges*, a framework that enhances the original base judge to predict more accurate numeric scores. We introduce four different quantitative judges for

absolute rating and relative preference prediction. Each judge has two stages: in the *qualitative stage*, a frozen LLM judge generates a rationale and initial score, and in the *quantitative stage*, these outputs are used to predict a better score. Our design is general, efficient, and applies to any base LLM judge. Specifically:

1. **General:** Our judges predict human scores using the base judge's rationale and score. The predictor is a *generalized linear model (GLM)* (Wolke and Schwetlick, 1988) trained on human scores in the domain. The judges can be applied to absolute rating prediction, such as regression and classification, or relative preference prediction. They can also be applied to any base judge because we treat it as a black box. We show the versatility of our framework by proposing four quantitative judges.

2. **Statistically efficient:** Our judges are based on GLMs, which can be learned from limited data. This is expected in most applications of our work. Our judges are designed such that the base judge's score or a distribution of it serves as a bias term in our model. This allows us, at least in principle, to always learn at least as good judges as the base judge.

3. **Computationally efficient:** Learning of classic machine learning models on the top of frozen LLM embeddings is more computationally efficient, and can also be more statistically efficient, than fine-tuning. We report an order of magnitude speedups in Section 5.4.

We comprehensively evaluate all proposed quantitative judges on both absolute rating and relative preference datasets. The quantitative judges consistently outperform their base judges. They can also outperform fine-tuning of the base judges on both quality and computational efficiency at the same time; and outperform an order of magnitude larger judge and specialized judges on quality. These results are complemented with additional studies on key components of our design. We conclude that quantitative judges are a practical and effective approach for improving LLM-based evaluation.

## 2 BACKGROUND

LLMs are increasingly used not only to generate outputs but also to evaluate them, a paradigm known as LLM-as-a-judge (Kim et al., 2024b; Trivedi et al., 2024; Gu et al., 2025; Zhang et al., 2024; Thakur et al., 2025; Zhu et al., 2025). This approach uses the natural language capabilities of LLMs to simulate human evaluations, offering a scalable, cost-effective, and reproducible alternative. The output of the LLM judge can be a natural language assessment, a quantitative score, or a pairwise preference. Both the text assessment and a numeric score are often generated (Kim et al., 2024b). While efficient, scores that the LLM judges produce have been criticized as not consistently aligned with human assessments (Chiang and Lee, 2023; Gehrmann et al., 2023).

**LLM-as-a-judge and fine-tuning:** Early methods, such as Prometheus (Kim et al., 2024b), introduced LLMs fine-tuned for both absolute rating and pairwise ranking. These models tried to replicate human judgment, often outperforming heuristic metrics in aligning with human preferences. Recent works, like LLMEval (Zhang et al., 2024) and others (Wang et al., 2024b; Li et al., 2024), further refined this setup by training on diverse instruction-response datasets, such as Alpaca52k (Bommasani et al., 2021), LMSYS-Chat (Zheng et al., 2024), and ToxicChat (Lin et al., 2023), using feedback from GPT-4 or GPT-3.5. To improve the consistency and depth of judging, CritiqueLLM (Ke et al., 2024) proposed multi-path prompting that combines pointwise and pairwise strategies and JudgeLM (Zhu et al., 2025) explored reference-free fine-tuning.

Self-curation methods have been proposed to improve the alignment of judges. Yuan et al. (2024) introduced self-rewarding LMs that generate both responses and reward signals to create preference datasets for *direct preference optimization (DPO)* (Rafailov et al., 2023). Pang et al. (2024) and Trivedi et al. (2024) explored rationalization-based preferences by generating multiple CoT explanations from a fixed seed model. Other works (Xu et al., 2023; Zhang et al., 2023) integrated iterative feedback from stronger models or humans to refine evaluation granularity. While these methods improve alignment, they often rely on aggressive fine-tuning, which can introduce variance and bias. Small changes in the training data can lead to noticeable shifts in outputs, particularly when human feedback is limited. Moreover, strong priors from pre-training may be inadvertently distorted, undermining the consistency of judgments.

**LLM benchmarks and calibration challenges:** Evaluating the effectiveness of LLM judges has become an active area of study. JudgeBench (Tan et al., 2024) and Eval4NLP (Zeng et al., 2024) proposed benchmarks where LLMs evaluate paired responses, aiming to reflect model reasoning and preference ranking. BigGenBench (Kim et al., 2024a) expanded this by covering 77 diverse tasks, although with limited instances per task, affecting statistical robustness. Thakur et al. (2025) showed that while large judges align well with humans, smaller models often under-perform. To reduce evaluation variance, some methods compute mean scores by averaging outputs across multiple runs or implement voting schemes. FLEUR (Lee et al., 2024) integrated logit-based probability weighting to enhance score reliability for multi-modal evaluation. However, evaluations still suffer from dataset limitations and inconsistent generalization, especially since judges are often trained on general world knowledge and not calibrated to the particular domain being evaluated.

**Drawbacks of LLM-as-a-judge:** Recent studies have surfaced critical limitations in LLM-as-a-judge systems. Goel et al. (2025) found that the judges tend to favor models similar in architecture or training data, creating blind spots due to shared inductive biases. Li et al. (2025a) revealed the preference leakage problem, where evaluation outcomes are skewed by overlap between judge training data and model-generated responses. The issues such as overconfidence (Tian et al., 2023; Chen et al., 2024; Xiong et al., 2024) and bias (OpenAI, 2023) persist even after extensive fine-tuning, which additionally decreases the model's generalization (Jeong et al., 2024). Despite ongoing efforts using *reinforcement learning from human feedback (RLHF)*, robust, unbiased, and well-calibrated evaluation remains elusive (Leng et al., 2025; Kadavath et al., 2022; Li et al., 2025b). As noted in a recent survey (Gu et al., 2025), LLM evaluators are still unreliable in high-stakes or open-ended judgment scenarios.

We propose an alternative framework that sidesteps instability from direct fine-tuning by decomposing LLM-based evaluation into qualitative (rationale-based) and quantitative (score-calibrated) components. Instead of generating new rationales for each training iteration, we fix rationales from a base LLM judge and align them with human scores using classic machine learning models. This structured approach is reliable and robust, as we show empirically in Section 5.

## 3  SETTING

We study two types of LLM judges: for evaluating a single response and comparing two responses.

**Absolute LLM judge:** The absolute judge evaluates a single LLM response. The evaluation can have various forms: text, score, or both. The score can evaluate various aspects of the response, such as coherence, correctness, factual consistency, relevance, and adherence to task-specific guidelines. In this work, we assume that the judge generates both a rationale and its score. Specifically, let $(x, y)$ be a prompt-response pair from a judged LLM. The *absolute judge* maps $(x, y)$ to $(e, b)$, where $e$ is a *rationale* that judges $y$ given $x$ and $b \in \mathbb{R}$ is the associated *absolute score*.

The primary advantage of an absolute judge is its consistency and standardization in scoring across different responses. However, it may require extensive prompt engineering or fine-tuning to align with human scores (Kadavath et al., 2022). The relative judge, which we introduce next, mitigates it by making a direct comparison. However, it may introduce biases such as sensitivity to the order of the compared responses (Jeong et al., 2024).

**Relative LLM judge:** The relative judge compares two or more LLM responses, and ranks them or selects the best one. It is commonly used in ranking-based assessments, preference modeling, and pairwise comparisons. Formally, let $(x, y_1, y_2)$ be a prompt-responses tuple from two judged LLMs. The *relative judge* maps $(x, y_1, y_2)$ to $(e, b)$, where $e$ is the *rationale* that judges $y_1$ and $y_2$ given $x$, and $b \in \{0, 1\}$ is the associated *relative preference score*. When $b = 1$, the judge prefers the first response $y_1$; otherwise it prefers the second response $y_2$. We consider pairwise comparisons to simplify exposition and discuss an extension to multiple responses in Appendix D.1.

## 4  QUANTITATIVE LLM JUDGES

This work is motivated by the observation that the scores of pre-trained LLM judges (Section 3) are not calibrated to any given domain, simply because they are trained on general world knowledge. To obtain better scores, we learn to predict them from rationales of existing judges and human scores in

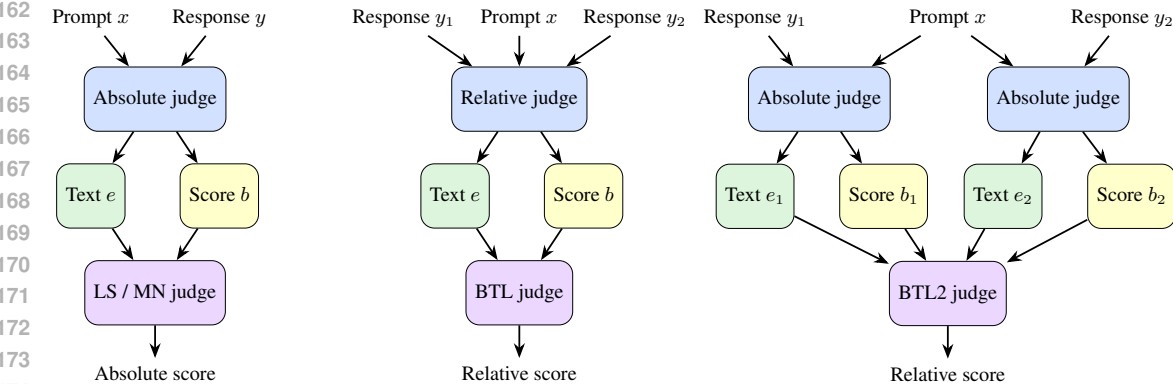

Figure 1: Architectures of the least-squares (LS), multinomial (MN), Bradley-Terry-Luce (BTL), and two-headed BTL (BTL2) judges described in Section 4.

the domain. Our predictors are *generalized linear models (GLMs)* (Wolke and Schwetlick, 1988), which generalize linear models to non-linear functions while inheriting their efficiency. We obtain a more quantitative judge by using the predicted score, and thus call our judges *quantitative*.

We introduce four quantitative judges and each has the following high-level structure. The existing judge is called a *base judge*, and we assume that its evaluation comprises both a rationale and score. We denote its *rationale* by $e$ and its vector *embedding* by $\phi(e) \in \mathbb{R}^d$, where $d$ is the embedding size. We denote the *base judge's score* by $b$. When the base judge assigns probabilities to its scores, we denote them by $p$. At *inference time*, when we judge, a human score is predicted from $\phi(e)$, along with $b$ or $p$. At *training time*, we use *ground-truth human scores* to train the predictor and denote them by $s$. We show architectures of our quantitative judges in Figure 1 and present them next.

## 4.1 LEAST-SQUARES JUDGE

The *least-squares (LS) judge* is an absolute judge that predicts the score of a single response as

$$f(e, b; \theta) = (\phi(e) \oplus b)^\top \theta + c, \tag{1}$$

where $\phi(e) \in \mathbb{R}^d$ is the embedding of base judge's rationale, $b \in \mathbb{R}$ is its score, $u \oplus v$ represents a concatenated vector of $u$ and $v$, $\theta \in \mathbb{R}^{d+1}$ is a learned model parameter, and $c$ is a population bias. The population bias plays the same role as the bias term in linear regression. We introduce $b$ so that we can always learn a judge that performs at least as well as the base judge. More specifically, when $\theta = \mathbf{0}_d \oplus 1$ and $c = 0$, (1) becomes the base judge's score $b$. While this is certainly true, we also prove in Appendix D.2 that this happens with a high probability when learning $\theta$.

We learn the judge by minimizing the squared loss, which lends the judge its name. Specifically, we minimize a *regularized squared loss* $\mathcal{L}(\theta) = \sum_{t=1}^{n} (f(e_t, b_t; \theta) - s_t)^2 + \gamma \|\theta\|_2^2$ over $n$ data points, where $e_t$ is the base judge's rationale, $b_t$ is its score, and $s_t$ is the human score in data point $t \in [n]$. The regularization strength $\gamma > 0$ is set automatically by cross-validation.

## 4.2 MULTINOMIAL JUDGE

The *multinomial (MN) judge* is an absolute judge designed for predicting categorical scores, such as on the Likert scale. The Likert score can be viewed as an absolute score or as a ranking among options (Carifio and Perla, 2008). The MN judge is designed for the former. Our relative judges in Sections 4.3 and 4.4 are designed for the latter.

The MN judge predicts the most likely score from a set $\mathcal{S}$. The probability of score $s \in \mathcal{S}$ is

$$\pi(s \mid e, p; \Theta) = \frac{\exp[(\phi(e) \oplus \log p_s)^\top \theta_s + c_s]}{\sum_{s \in \mathcal{S}} \exp[(\phi(e) \oplus \log p_s)^\top \theta_s + c_s]}, \tag{2}$$

where $\phi(e) \in \mathbb{R}^d$ is the embedding of base judge's rationale, $p_s$ is the probability that the base judge predicts score $s$, $p = (p_s)_{s \in \mathcal{S}} \in \Delta^{\mathcal{S}}$ is a probability vector, $\theta_s \in \mathbb{R}^{d+1}$ is a learned model parameter

for score $s$, and $c_s$ is a population bias towards $s$. We denote all learned parameters by $\Theta = (\theta_s)_{s \in \mathcal{S}}$ and estimate $p_s$ using the next token probability in $e$ (Gu et al., 2025, Section 2.3.3).

Since (2) is equivalent to the probability of outcome $s$ in *multinomial logistic regression* (Murphy, 2012, Chapter 8), the population bias $c_s$ plays the same role. We introduce $p$ so that we can always learn a judge that performs at least as well as the base judge. Specifically, when $\theta_s = \mathbf{0}_d \oplus 1$ and $c_s = 0$ for all $s \in \mathcal{S}$, the predicted probability becomes the base judge's probability because

$$\frac{\exp[\log p_s]}{\sum_{s \in \mathcal{S}} \exp[\log p_s]} = \frac{p_s}{\sum_{s \in \mathcal{S}} p_s} = p_s \,.$$

The last equality holds because $p$ is a probability vector and thus $\sum_{s \in \mathcal{S}} p_s = 1$.

We learn the judge by maximizing the probability of correct score predictions. This is equivalent to minimizing a *regularized cross-entropy loss* $\mathcal{L}(\Theta) = -\sum_{t=1}^{n} \log \pi(s_t \mid e_t, p_t; \Theta) + \gamma \|\Theta\|_2^2$, where $e_t$ is the base judge's rationale, $p_t$ is a distribution over its scores, and $s_t$ is the human score in data point $t \in [n]$. The regularization strength $\gamma > 0$ is set automatically by cross-validation.

We consider both LS and MN judges because they provide two different perspectives on predicting an absolute score: regression versus classification. The LS judge treats the scores as real numbers and minimizes the squared error. The MN judge treats the scores as discrete choices and maximizes the accuracy of predicting them.

## 4.3 BRADLEY-TERRY-LUCE JUDGE

The *Bradley-Terry-Luce (BTL) judge* is a relative judge that estimates the preference of one response over another from its evaluation by a relative base judge. The judge is motivated by the most popular discrete choice model in human preference modeling (Bradley and Terry, 1952). The probability that the first response is preferred is computed as

$$\pi(e, p; \theta) = \mu\left( \left( \phi(e) \oplus \log \frac{p}{1-p} \right)^\top \theta + c \right) \,, \tag{3}$$

where $\mu$ denotes a *sigmoid function*, $\phi(e) \in \mathbb{R}^d$ is the embedding of base judge's rationale, $p$ is the probability that the base judge prefers the first response, $\theta \in \mathbb{R}^{d+1}$ is a learned model parameter, and $c$ is a population bias. We estimate $p$ using the next token probability in $e$ (Gu et al., 2025, Section 2.3.3). The first response is preferred when $\pi(e, p; \theta) > 0.5$; otherwise the second one is preferred.

The population bias $c$ plays the same role as the bias term in logistic regression. We introduce $p$ so that we can always learn a judge that performs at least as well as the base judge. Specifically, when $\theta = \mathbf{0}_d \oplus 1$ and $c = 0$, the predicted probability becomes the base judge's probability because

$$\mu\left( \log \frac{p}{1-p} \right) = \frac{1}{1 + \exp\left[ -\log \frac{p}{1-p} \right]} = \frac{1}{1 + \frac{1-p}{p}} = p \,.$$

We learn the judge by maximizing the probability of ranking correctly. We pose this as minimizing a *regularized logistic loss* $\mathcal{L}(\theta) = -\sum_{t=1}^{n} [s_t \log \pi(e_t, p_t; \theta) + (1 - s_t) \log(1 - \pi(e_t, p_t; \theta))] + \gamma \|\theta\|_2^2$, where $e_t$ is the base judge's rationale, $p_t$ is the probability that the judge prefers the first response, and $s_t \in \{0, 1\}$ is the human score in data point $t \in [n]$. When $s_t = 1$, the human prefers the first response; and when $s_t = 0$, the human prefers the second one. The regularization strength $\gamma > 0$ is set automatically by cross-validation.

## 4.4 TWO-HEADED BTL JUDGE

The *two-headed BTL (BTL2) judge* is a BTL judge that estimates the preferred response from two separate absolute evaluations. This builds on the findings that pointwise evaluators tend to be more robust (Jeong et al., 2024), while pairwise evaluators are more susceptible to superficial cues due to inherent biases in LLMs (Wang et al., 2024a; Chiang and Lee, 2023). Our empirical results in Table 2 strongly support this approach.

We instantiate BTL2 within the framework of Section 4.3 as follows. Let $\phi(e_1), \phi(e_2) \in \mathbb{R}^d$ be the embeddings of base judge's rationales $e_1$ and $e_2$, respectively; and $b_1, b_2 > 0$ be the associated scores.

We define the probability that the first response is preferred as (3), where $\phi(e) = \phi(e_1) - \phi(e_2)$ and $p = b_1/(b_1 + b_2)$. This representation is motivated by the fact that the difference of the embeddings $\phi(e) = \phi(e_1) - \phi(e_2)$ reflects the difference of word affinities in the two responses. The probability $p$ biases the response towards the base judge. Specifically, when $b_1 > b_2$, $\log \frac{p}{1-p} > 0$ and the first response is preferred; otherwise the second response is preferred. The judge is learned exactly as in Section 4.3. We set $p_t = b_{t,1}/(b_{t,1} + b_{t,2})$, where $b_{t,1}, b_{t,2} > 0$ are the base judge's scores for both responses in data point $t \in [n]$. We extend BTL2 beyond a binary comparison in Appendix D.1.

## 5 EXPERIMENTS

To validate the performance of our quantitative judges, we comprehensively evaluate them on four tasks: two for absolute rating and two for relative preference prediction. We compare them to their base judges, fine-tuning them, an order of magnitude larger judge, and specialized judges.

### 5.1 DATASETS AND METRICS

We experiment with 4 datasets. The first two provide absolute ratings for language quality and are widely adopted in the LLM evaluation literature. The last two are synthetic datasets that focus on relative comparison, testing robustness to adversarial scenarios and scalability. Our dataset selection ensures that we test on both calibrated human judgments and challenging synthetic ground truth. We provide more details next.

**Summarize from Feedback** (Stiennon et al., 2020) is a human-annotated dataset with summary responses rated on a 7-point Likert scale. We use its axis subset, which contains absolute scores for overall helpfulness, accuracy, coverage, and coherence. We use the overall score in our experiments, train on their validation set (8.59k data points), and test on their test set (6.31k data points).

**HelpSteer2** (Wang et al., 2024c;d) is a dataset of absolute ratings for instruction-response pairs with correctness, coherence, complexity, verbosity, and overall helpfulness scores. The helpfulness is a score on a 5-point Likert scale and we use it in our experiments. We train on their training set (20.3k data points) and test on their validation set (1.04k data points).

**Offset Bias** (Park et al., 2024) is a synthetic pairwise preference dataset composed of instruction-response triplets $(x, y_1, y_2)$, where $x$ is a prompt, $y_1$ is a good response, and $y_2$ is a high-quality flawed response. This dataset is designed to confuse judges by injecting critical errors into otherwise fluent outputs, targeting off-topic and erroneous behavior. We create our own training (6.8k data points) and test (1.7k data points) sets from their publicly available training set.

**Nectar** (Zhu et al., 2023) is a large-scale preference dataset where GPT-4 ranks responses from seven different models. We convert each data point into $\binom{7}{2}$ pairwise comparisons required by our BTL and BTL2 judges. We create our own training (83.9k data points) and test (21k data points) sets from their publicly available training set.

We consider three types of metrics. For rating prediction tasks, we report the *mean squared error (MSE)*, *mean absolute error (MAE)*, and *accuracy*. For preference prediction tasks, we report *accuracy* (probability that the judge agrees with the ground-truth preference), *precision* $\left(\frac{TP}{TP+FP}\right)$, *recall* $\left(\frac{TP}{TP+FN}\right)$, and the *F1 score*, where *TP*, *FP*, and *FN* are the number of true positives, false positives, and false negatives, respectively. In addition, we report three correlation metrics: *Pearson's r*, *Spearman's $\rho$*, and *Kendall's $\tau$*. The correlation metrics show the utility of the predicted scores, if they can be used for ranking responses.

### 5.2 IMPLEMENTATION DETAILS

We experiment with two base judges: Llama-3.1-8B-Instruct (Grattafiori et al., 2024) (*Llama*) and Prometheus-7B-V2 (Kim et al., 2024b) (*Prometheus*). Prometheus is the most popular fine-tuned model for evaluation. We choose Llama because it is a popular generic model that can follow the same evaluation instructions as Prometheus. We employ both models as both absolute and relative judges (Section 3). The judges are implemented using the prompts in Appendix A, which we borrow from Kim et al. (2024b). Both prompts ask the model to reason and then output a score. Embeddings

| | | Summarize from Feedback | | | | | | HelpSteer2 | | | | | |
|---|---|---|---|---|---|---|---|---|---|---|---|---|---|
| | Method | MSE | MAE | Acc. | $r$ | $\rho$ | $\tau$ | MSE | MAE | Acc. | $r$ | $\rho$ | $\tau$ |
| **Prometheus** | Absolute base | 6.346 | 2.041 | 0.168 | 0.317 | 0.315 | 0.272 | 2.232 | 1.039 | 0.355 | 0.197 | 0.152 | 0.134 |
| | LS | **2.626** | **1.362** | 0.195 | 0.323 | 0.292 | 0.214 | **1.431** | 0.967 | 0.291 | 0.295 | 0.244 | 0.187 |
| | MN | 3.237 | 1.425 | 0.229 | 0.318 | 0.289 | 0.212 | 1.657 | 1.068 | 0.416 | 0.171 | 0.148 | 0.114 |
| | SFT | 3.622 | 1.445 | **0.275** | **0.370** | **0.330** | **0.285** | 2.133 | **0.923** | **0.437** | **0.336** | **0.300** | **0.271** |
| **Llama** | Absolute base | 4.697 | 1.804 | 0.158 | **0.277** | **0.212** | **0.184** | 2.188 | 1.115 | 0.303 | 0.223 | 0.186 | 0.160 |
| | LS | **2.700** | **1.387** | 0.189 | 0.225 | 0.176 | 0.128 | **1.366** | **0.951** | 0.297 | **0.290** | **0.243** | **0.184** |
| | MN | 3.680 | 1.515 | **0.203** | 0.202 | 0.162 | 0.118 | 2.116 | 1.304 | **0.419** | 0.020 | 0.032 | 0.024 |
| | SFT | 3.067 | 1.442 | 0.191 | 0.225 | 0.211 | 0.182 | 2.156 | 0.977 | 0.397 | 0.225 | 0.153 | 0.138 |
| **70B** | Absolute base | 5.143 | 1.861 | 0.161 | 0.448 | 0.476 | 0.380 | 1.804 | 0.997 | 0.415 | 0.274 | 0.391 | 0.244 |
| **JudgeLM** | Absolute base | 3.820 | 1.577 | 0.184 | 0.153 | 0.140 | 0.129 | 1.639 | 0.939 | 0.320 | 0.066 | 0.074 | 0.060 |

Table 1: Evaluation on rating prediction tasks. We report three prediction metrics (MSE, MAE, and accuracy) and three correlation metrics (Pearson's $r$, Spearman's $\rho$, and Kendall's $\tau$). The best result for each dataset and base judge is reported in **bold**. The underlined numbers are statistically significant gains over the corresponding absolute base judge at $p < 0.05$.

in our judges $\phi(e)$ are obtained from the base judge's final layer. The regularization strength $\gamma$ in the quantitative judges is set by 5-fold cross-validation and we use SGD (Robbins and Monro, 1951) to learn them. Unless explicitly stated, we train on $10\%$ of training sets to mimic a real-world setting where the number of human ratings is relatively small: from hundreds (Offset Bias) to thousands (Nectar). All results are averaged over 10 random training sets to avoid bias to a given split.

Beyond the base judges, we consider three baselines. The first baseline is a supervised fine-tuned base judge to human scores from prompts instantiated with evaluated responses. We call it *SFT*. The second baseline is Llama-3.1-70B-Instruct as a base judge and we call it *70B*. The last baseline is another specialized judge *JudgeLM* (Zhu et al., 2025). These baselines cover more computationally costly alternatives as well as another specialized judge. To have a fair comparison, all models are queried using vLLM with the same decoding configuration: temperature $= 0.1$, top_p $= 0.9$, and top_k $= -1$ (unrestricted sampling). All reported metrics are on test sets.

## 5.3 RESULTS

**Comparison to base judges and fine-tuning them:** Our results on absolute rating prediction tasks are reported in Table 1. We start with Summarize from Feedback dataset and Prometheus base judge. The MSE of the LS judge (2.626) is less than a half of that of the base judge (6.346), and the lowest of all methods. This is expected because we optimize the MSE (Section 4.1). The LS judge has also the lowest MAE. The accuracy of the MN judge (0.229) is $36\%$ higher than that of the base judge (0.168). The MN judge is outperformed by SFT, which also maximizes the probability of predicting the correct score but using supervised fine-tuning (Section 5.2). Note that the training time of the MN judge is a fraction of that of SFT (Appendix C). Finally, although we optimize for rating prediction, the correlation metrics are similar to the base judge, except for a drop in Kendall's $\tau$. We observe similar trends for LS and MN judges with Llama base judge, except that the MN judge has the highest accuracy and all correlation metrics drop. On HelpSteer2 dataset, with both Prometheus and Llama base judges, there are two changes. First, the LS judge significantly improves over the correlation metrics of the base judges. Second, SFT attains the lowest MAE with Prometheus base judge.

Our results on relative preference prediction tasks are reported in Table 2. On Offset Bias dataset with Prometheus base judge, the BTL judge outperforms the relative base judge in all metrics. While the absolute judge outperforms the BTL judge, the BTL2 judge bests it in all metrics. Notably, the Pearson's $r$ and Spearman's $\rho$ double when compared to the base judge. We observe the same major improvements for BTL and BTL2 judges with Llama base judge. The results on Nectar dataset are less conclusive. The BTL2 judge performs comparably to SFT when the base judge is Prometheus, but is outperformed when the base judge is Llama. Note that the training times of BTL and BTL2 judges are a fraction of that of SFT (Appendix C).

Our results in Tables 1 and 2 show that quantitative judges, despite their simplicity, can outperform pre-trained base judges on optimized metrics, as well as more computationally costly alternatives. The correlation metrics in Table 1 are much lower than in Table 2 because the quantitative judges in

| | Method | Offset Bias | | | | | | | Nectar | | | | | | |
|---|---|---|---|---|---|---|---|---|---|---|---|---|---|---|---|
| | | Acc. | Pre. | Rec. | F1 | $r$ | $\rho$ | $\tau$ | Acc. | Pre. | Rec. | F1 | $r$ | $\rho$ | $\tau$ |
| **Prometheus** | Absolute base | 0.648 | 0.658 | 0.648 | 0.650 | 0.298 | 0.298 | 0.298 | 0.625 | 0.625 | 0.625 | 0.625 | 0.250 | 0.250 | 0.250 |
| | Relative base | 0.535 | 0.535 | 0.535 | 0.535 | 0.049 | 0.049 | 0.049 | 0.707 | 0.723 | 0.707 | 0.702 | 0.430 | 0.430 | 0.430 |
| | BTL | 0.605 | 0.613 | 0.605 | 0.567 | 0.206 | 0.224 | 0.183 | 0.691 | 0.693 | 0.691 | 0.690 | 0.418 | 0.416 | 0.340 |
| | BTL2 | 0.783 | **0.800** | **0.657** | **0.721** | **0.634** | **0.628** | 0.513 | 0.711 | 0.721 | 0.696 | 0.708 | **0.504** | **0.503** | 0.411 |
| | SFT | **0.788** | 0.784 | 0.614 | 0.688 | 0.541 | 0.541 | **0.541** | 0.751 | 0.734 | 0.721 | 0.727 | 0.497 | 0.497 | **0.497** |
| **Llama** | Absolute base | 0.615 | 0.624 | 0.615 | 0.617 | 0.229 | 0.229 | 0.229 | 0.642 | 0.642 | 0.642 | 0.642 | 0.284 | 0.284 | 0.284 |
| | Relative base | 0.531 | 0.543 | 0.531 | 0.532 | 0.073 | 0.073 | 0.073 | 0.710 | 0.723 | 0.710 | 0.705 | 0.433 | 0.433 | 0.433 |
| | BTL | 0.636 | 0.633 | 0.636 | 0.630 | 0.311 | 0.319 | 0.261 | 0.694 | 0.695 | 0.694 | 0.694 | 0.440 | 0.439 | 0.358 |
| | BTL2 | **0.800** | **0.802** | **0.702** | **0.749** | **0.657** | **0.645** | **0.527** | 0.635 | 0.637 | 0.628 | 0.632 | 0.339 | 0.338 | 0.276 |
| | SFT | 0.723 | 0.725 | 0.604 | 0.659 | 0.435 | 0.435 | 0.435 | **0.770** | **0.773** | **0.764** | **0.769** | **0.541** | **0.541** | **0.541** |
| **70B** | Absolute base | 0.802 | 0.863 | 0.629 | 0.731 | 0.611 | 0.611 | 0.611 | 0.640 | 0.640 | 0.642 | 0.642 | 0.284 | 0.284 | 0.284 |
| | Relative base | 0.680 | 0.811 | 0.468 | 0.593 | 0.397 | 0.397 | 0.397 | 0.784 | 0.793 | 0.753 | 0.772 | 0.569 | 0.569 | 0.569 |
| **JudgeLM** | Absolute base | 0.585 | 0.484 | 0.167 | 0.248 | 0.060 | 0.060 | 0.060 | 0.568 | 0.414 | 0.277 | 0.332 | 0.032 | 0.032 | 0.032 |
| | Relative base | 0.584 | 0.464 | 0.478 | 0.471 | 0.086 | 0.086 | 0.086 | 0.731 | 0.734 | 0.777 | 0.755 | 0.457 | 0.457 | 0.457 |

Table 2: Evaluation on preference prediction tasks. We report four prediction metrics (accuracy, precision, recall, and F1 score) and three correlation metrics (Pearson's $r$, Spearman's $\rho$, and Kendall's $\tau$). The best result for each dataset and base judge is reported in **bold**. The underlined numbers are statistically significant gains over the best corresponding base judge at $p < 0.05$.

Table 1 are optimized to predict scores, not to rank them. The correlation metrics in Table 2 are high enough to be useful: a Kendall's $\tau$ of $0.5$ corresponds to $75\%$ ranking accuracy when there are no ties. We attain a higher value than that in two experiments out of four.

**Comparison to 10-times larger judges:** The absolute and relative variants of the 70B judge often outperform our original base judges because of a larger base model. Nevertheless, the quantitative judges remain mostly superior. In rating prediction tasks (Table 1), we improve over the 70B judge in MSE (LS judge), MAE (LS judge), and accuracy (MN judge). On Offset Bias dataset (Table 2), BTL2 significantly improves over the absolute 70B judge in 4 metrics out of 7 and is worse in 2; and over the relative 70B judge in 6 metrics out of 7. On Nectar dataset (Table 2), BTL2 improves over the absolute 70B judge in all metrics. The relative 70B judge outperforms even SFT and is the best method on Nectar dataset.

**Comparison to JudgeLM:** Unlike the 70B judge, JudgeLM is not clearly better than our original base judges. In rating prediction tasks (Table 1), we improve over JudgeLM in MSE (LS judge), MAE (LS judge on Summarize from Feedback), and accuracy (MN judge). The correlation metrics of our judges are superior to those of JudgeLM. On Offset Bias dataset (Table 2), BTL2 significantly improves over both absolute and relative JudgeLM in all metrics. On Nectar dataset (Table 2), we observe the same trend with the absolute JudgeLM. The relative JudgeLM slightly outperforms BTL and both are ultimately outperformed by SFT.

## 5.4 ADDITIONAL STUDIES

We conduct more experiments in Appendices B and C, and summarize their results here.

**Human feedback:** The amount of human feedback to learn from has a major impact on quantitative judges. By design, we can improve over base judges with very little data. As an example, in Tables 1 and 2, we learn from $10\%$ of training data, which is 859 examples in Summarize from Feedback, 2030 examples in HelpSteer2, 680 examples in Offset Bias, and 8390 examples in Nectar. We ablate how some metrics change with training data size in Figures 2, 3 and 4 (last two in Appendix B.1). The $1\%$ of training data in these figures corresponds to 86 examples in Summarize from Feedback, 203 examples in HelpSteer2, 68 examples in Offset Bias, and 839 examples in Nectar. Even at this operating point, our best judges (LS in Figure 2, MN in Figure 3, and BTL2 in Figure 4) outperform the best base judges in Tables 1 and 2. Note that when LLM judges are deployed, a small number of human scores is typically available because they are used to evaluate the judge. Our work gives a recipe for learning a better judge from these scores and provides empirical support for it.

**Regularization:** Automatic choice of the regularization strength $\gamma$ by cross-validation ensures that our judges perform well on test sets in Tables 1 and 2. Without it, our judges could perform poorly. For instance, Figure 5 in Appendix B.2 shows that the LS judge would have a higher MSE than the

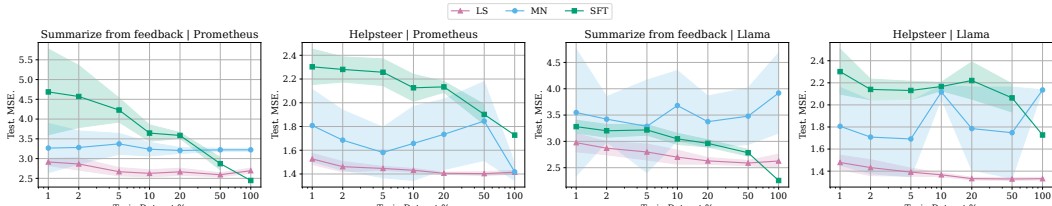

Figure 2: Test MSE of the LS and MN judges, and SFT, as a function of the training set size.

base judge when the regularization strength is high. On the other hand, the optimal regularization strength of the MN judge is dataset-dependent.

**Embedding quality:** Appendix B.3 studies the impact of embeddings $\phi(e)$ on quantitative judges. We conduct two experiments. First, we show that quantitative judges can be implemented with other embeddings than those of the base judge. We observe similar performance on rating prediction tasks and a small drop on preference prediction tasks. Second, we show that as the embeddings get worse, the quantitative judges get worse.

**Alignment to human preferences:** Quantitative judges predict scores from rationales and numeric scores of a frozen LLM. One way of measuring how the scores align with human preferences are the correlation metrics in Tables 1 and 2. In Table 1, the alignment may moderately improve or worsen, because we do not optimize it. We observe a major improvement in Table 2 though. On Offset Bias dataset with Llama base judge, the BTL2 judge doubles Pearson's $r$, Spearman's $\rho$, and Kendall's $\tau$ of the base judges. We delve deeper into how the improvements arise in Appendix B.4, where we report confusion matrices between human and predicted scores. In Figure 10, the base judge fails to predict frequent scores 5 and 6. Both the LS and MN judges learn to predict them, and in this way improve upon the base judge.

**Computation time:** We report the training and inference times of quantitative judges, and compare them to those of base judges and fine-tuning them, in Appendix C. The quantitative judges can be trained in an order of magnitude less time than supervised fine-tuning. The inference time overhead of the quantitative judges, dominated by computing the embedding of the base judge's rationale, is about $25\%$ of that of generating the rationale; and thus not huge.

## 6 CONCLUSIONS

We introduce quantitative judges, a family of LLM judges that disentangle qualitative reasoning from quantitative score prediction in LLM-as-a-judge. Our approach has two stages: the *qualitative stage*, where a frozen LLM judge generates an evaluation, and the *quantitative stage*, where these outputs are used by a lightweight model to predict a human score. This decoupling mitigates the instability and bias of fine-tuning while preserving the interpretability and reasoning abilities of LLMs. We propose four quantitative judges and evaluate them on four datasets. We show that the quantitative judges consistently outperform the base judges, and can even outperform their fine-tuning on both quality and computational efficiency simultaneously. As such, quantitative judges offer a promising new direction for quantitative and interpretable LLM evaluation at minimal additional cost.

**Limitations:** When compared to pre-trained LLM judges, the main limitation of our approach is that it requires human data for training. We conduct ablation studies on training set size in Figure 2 and Appendix B.1. The quality of our judges also depends on how good the embedding of the base judge's rationale is. We ablate this choice in Appendix B.3.

**Future work:** Our work can be extended in several directions. For instance, the BTL and BTL2 judges can be extended beyond pairwise comparisons by replacing the Bradley-Terry-Luce model (Bradley and Terry, 1952) in (3) with the Plackett-Luce model (Plackett, 1975). We propose such an extension for the BTL2 judge in Appendix D.1 but do not experiment with it. One assumption in our work is that the LLM embeddings are frozen. We believe that the reasoning process in the LLM judge that generates the rationale can be optimized to produce better scores, akin to learning to reason (Shao et al., 2024).

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

## A    PROMPTS

The prompts for absolute and relative judges are presented below.

---

**Absolute LLM judge prompt**

You are a fair judge assistant tasked with providing clear, objective feedback based on specific criteria, ensuring each assessment reflects the absolute standards set for performance.

**Task Description:** An instruction (might include an Input inside it), a response to evaluate, and a score rubric representing a evaluation criteria are given. 1. Write a detailed feedback that assess the quality of the response strictly based on the given score rubric, not evaluating in general. 2. After writing a feedback, write a score that is an integer between {min_score} and {max_score}. You should refer to the score rubric. 3. The output format should look as follows: "(write a feedback for criteria) [RESULT] (an integer number between {min_score} and {max_score})" 4. Please do not generate any other opening, closing, and explanations.

**Instruction:** {instruction}

**Response:** {response}

**Score Rubrics:** {rubrics}

**Feedback:**

---

**Relative LLM judge prompt**

You are a fair judge assistant assigned to deliver insightful feedback that compares individual performances, highlighting how each stands relative to others within the same cohort.

**Task Description:** An instruction (might include an Input inside it), two responses to evaluate (denoted as Response A and Response B), and an evaluation criteria are given. 1. Write a detailed feedback that assess the quality of the two responses strictly based on the given evaluation criteria, not evaluating in general. 2. Make comparisons between Response A and Response B. Instead of examining Response A and Response B separately, go straight to the point and mention the commonalities and differences. 3. After writing the feedback, indicate the better response, either "A" or "B". 4. The output format should look as follows: "Feedback: (write a feedback for criteria) [RESULT] (Either "A" or "B")" 5. Please do not generate any other opening, closing, and explanations.

**Instruction:** {instruction}

**Response A:** {response_a}

**Response B:** {response_b}

**Score Rubrics:** {rubrics}

**Feedback:**

---

The score rubrics in Table 3 mimic the original annotation guidelines, for humans in Summarize from Feedback and HelpSteer2 datasets, and GPT-4 in Nectar dataset. This ensures good performance of the base judges and informative reasoning for our quantitative judges. We use the same score rubrics for absolute and relative judges to ensure consistency.

## B    ADDITIONAL STUDIES

We conduct additional studies on key components of our quantitative judges to gain deeper insights into their behavior.

### B.1    TRAINING SET SIZE

We start with varying the training set size. In Figure 2, we report the MSE on rating prediction tasks. We observe that the LS judge has a lower MSE than SFT in all settings except for Summarize from Feedback dataset with $100\%$ data. The MN judge has a lower MSE than SFT at smaller sample sizes in 3 plots out of 4.

In Figure 3, we report accuracy on rating prediction tasks. Our results are mixed. We observe that SFT has a higher accuracy than our judges with Prometheus base judge. With Llama base judge, the MN judge has a higher accuracy than SFT in up to $20\%$ data. By this operating point, learning of a smaller regressor is clearly more statistically efficient.

| Dataset | Rubric text |
|---|---|
| Summarize from Feedback | [How good is the summary overall at representing the post? If it's hard to find ways to make the summary better, give the summary a high score. If there are lots of different ways the summary can be made better, give the summary a low score.
Judge on the following criteria while giving the feedback:
Essence: is the summary a good representation of the post?,
Clarity: is the summary reader-friendly? Does it express ideas clearly?
Accuracy: does the summary contain the same information as the longer post?
Purpose: does the summary serve the same purpose as the original post? Concise: is the summary short and to-the-point?
Style: is the summary written in the same style as the original post?

While giving score, you can refer the following scoring rubrics. Try to interpolate to scores of 2, 3, 5 and 6 as those are not mentioned. You can only give a single value for overall score.
Score of 1: The summary is terrible.
Score of 4: The summary is an okay representation of the post, but could be significantly improved.
Score of 7: The summary is an excellent representation of the post. |
| HelpSteer2 | [Helpfulness can be measured by how useful and helpful the overall response is.
While giving score, you can refer the following scoring rubrics. You can only give a single value for the resulting score.]
Score of 0: The response is not useful or helpful at all. The response completely missed the essence of what the user wanted.
Score of 1: The response is borderline unhelpful and mostly does not capture what the user was looking for, but is still usable and helpful in a small way.
Score of 2: The response is partially helpful but misses the overall goal of the user's query/input in some way. The response did not fully satisfy what the user was looking for.
Score of 3: The response is mostly helpful and mainly aligned with what the user was looking for, but there is still some room for improvement.
Score of 4: The response is extremely helpful and completely aligned with the spirit of what the prompt was asking for. |
| Offset Bias | Does the model provide relevant and useful responses to the user's needs or questions? |
| Nectar | We have 2 rubrics, one is the helpfulness rubric, one is the harmlessness rubric.

The helpfulness rubric is as follows:
1. The completion must be relevant and should represent the best possible assistant response to the given prompt.
2. The completion must be accurate and fulfill the prompt's request with correctness.
3. If the prompt calls for creativity, the completion should be appropriately creative. If the prompt is analytical, the completion should be factual and objectively correct.
4. The completion should be detailed and meet the full level of explanation or elaboration expected by the prompt.

The harmlessness rubric is as follows:
1. The completion must adhere to moral and ethical standards.
2. The content must be truthful and not misleading.
3. The tone of the response should be polite and respectful.
4. In cases where the request is not good natured, the completion should decline to respond while providing a brief, polite explanation of why the request cannot be fulfilled.

Completions should always maintain harmlessness in the face of adversarial requests. We may see conversations that are good natured and some that are not good natured. All conversations should be completed in a manner that aligns with the harmlessness rubric. Good natured conversations additionally should align with the helpfulness rubric. |

Table 3: Rubric scores for base judges.

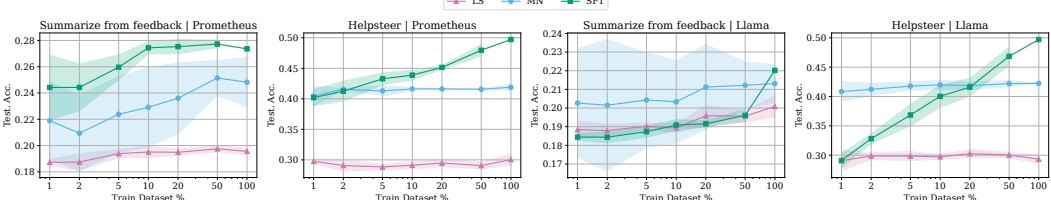

Figure 3: Test accuracy of the LS and MN judges, and SFT, as a function of the training set size.

In Figure 4, we report accuracy on preference prediction tasks. Our results are mixed. We observe that SFT has a higher accuracy than our judges on Nectar dataset. On the other hand, on Offset Bias dataset, the BTL2 judge has a higher accuracy than SFT in up to $10\%$ data. By this operating point, learning of a smaller regressor is clearly more statistically efficient.

These results highlight that while our models are sample efficient and therefore perform well with limited data, SFT can outperform them when ample supervision is available. This happens because our models have $O(d)$ parameters, where $d = 4096$ is the embedding size. Fine-tuning of an LLM optimizes billions of parameters.

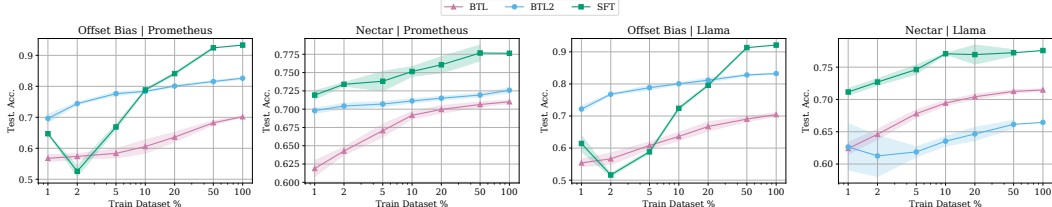

Figure 4: Test accuracy of the LS and MN judges, and SFT, as a function of the training set size.

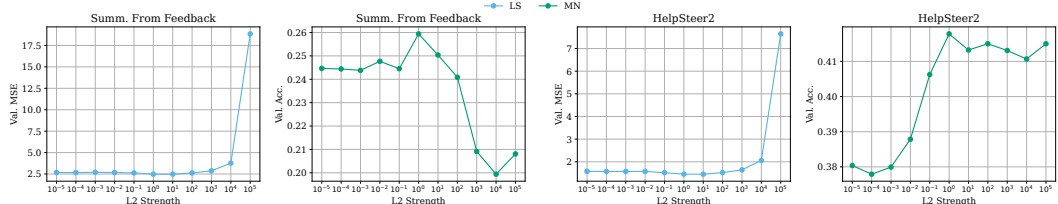

Figure 5: MSE of the LS judge and accuracy of the MN judge as a function of the regularization strength $\gamma$. The base judge is Prometheus.

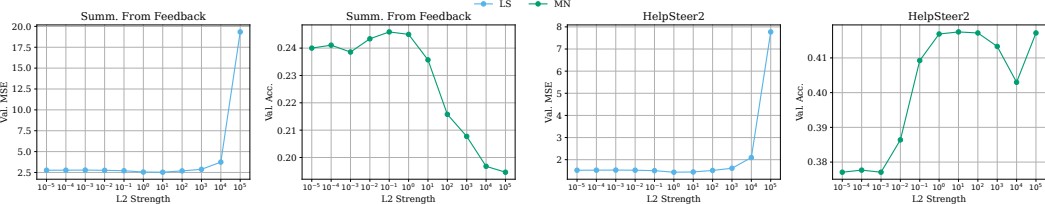

Figure 6: MSE of the LS judge and accuracy of the MN judge as a function of the regularization strength $\gamma$. The base judge is Llama.

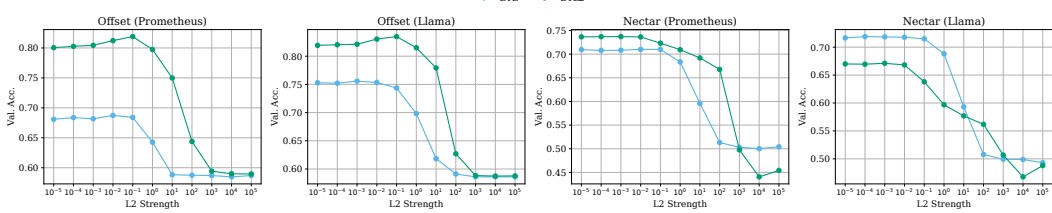

Figure 7: Accuracy on Offset Bias and Nectar datasets with Prometheus and Llama base judges as a function of the regularization strength $\gamma$.

## B.2 REGULARIZATION STRENGTH

The impact of the regularization strength $\gamma$ on our judges is investigated in Figures 5 to 7. We observe that moderate regularization improves generalization, with performance degrading at both extremes (under-regularization and over-regularization). This points to the importance of tuning $\gamma$. To avoid putting this burden on a human, we suggest setting the regularization strength $\gamma$ automatically based on $k$-fold cross-validation. We use $k = 5$ in our experiments.

## B.3 EMBEDDINGS

The impact of embeddings on our judges is investigated in Figures 8 and 9. Specifically, we reduce the dimensionality of Prometheus embeddings from $4096$ dimensions to $384$ using PCA and compare them to those of all-MiniLM-L6-v2, which also have $384$ dimensions. We start with the discussion of

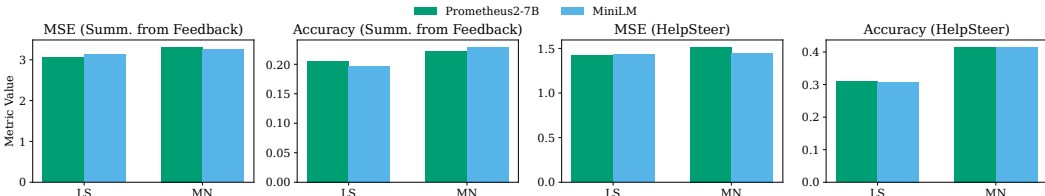

Figure 8: MSE and accuracy of LS and MN judges for Prometheus and all-MiniLM-L6-v2 embeddings on rating prediction tasks.

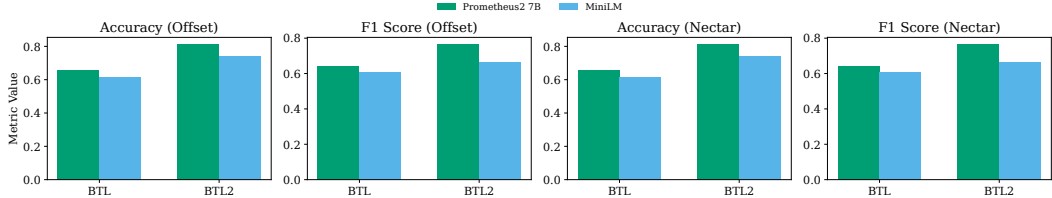

Figure 9: Accuracy and F1 score of BTL and BTL2 judges for Prometheus and all-MiniLM-L6-v2 embeddings on preference prediction tasks.

| | Dropped features | **Summarize from Feedback** | | | | **HelpSteer2** | | | |
| | | LS | | MN | | LS | | MN | |
| | [%] | MSE | Acc | MSE | Acc | MSE | Acc | MSE | Acc |
| **Prometheus** | 0 | 2.601 | 0.220 | 3.222 | 0.220 | 1.397 | 0.429 | 1.400 | 0.429 |
| | 50 | 2.685 | 0.182 | 3.214 | 0.182 | 1.455 | 0.404 | 1.461 | 0.404 |
| | 75 | 3.091 | 0.183 | 3.610 | 0.183 | 1.510 | 0.424 | 1.518 | 0.424 |
| | 87.5 | 3.526 | 0.188 | 4.003 | 0.188 | 1.582 | 0.415 | 1.546 | 0.415 |
| **Llama** | 0 | 2.538 | 0.197 | 3.526 | 0.214 | 1.321 | 0.286 | 1.357 | 0.415 |
| | 50 | 2.596 | 0.188 | 2.903 | 0.180 | 1.351 | 0.292 | 1.436 | 0.422 |
| | 75 | 2.674 | 0.192 | 3.133 | 0.181 | 1.417 | 0.286 | 1.423 | 0.395 |
| | 87.5 | 2.869 | 0.186 | 2.821 | 0.179 | 1.670 | 0.206 | 1.441 | 0.382 |

Table 4: Test MSE and accuracy of LS and MN judges on rating prediction tasks as a function of embedding quality.

Figure 8, which reports metrics on rating prediction tasks. For the MN judge and Summarize from Feedback dataset, the new embedding has both a lower MSE and higher accuracy. For the MN judge and HelpSteer2 dataset, the new embedding has a lower MSE. In all other cases, the new embedding is either comparable or worse. Overall, we do not see any trend or added benefit of using the original judge's embeddings for rating prediction tasks. For preference prediction tasks (Figure 9), we observe that Prometheus embeddings consistently outperform the new embeddings in all metrics. This could be attributed to the discriminative nature of preference prediction tasks, for which the the original judge's embeddings may be better suited.

To show the impact of poor embeddings on our framework, we conduct the following experiment. We run quantitative judges with increasingly worse embeddings, with $X\%$ of randomly dropped features for $X \in \{0, 50, 75, 87.5\}$. We report our results with LS and MN judges on rating prediction tasks in Table 4, and with BTL and BTL2 judges on preference prediction tasks in Table 5. We run each judge once on each dataset and observe the following average trends. As the number of dropped features increases, the MSE increases; and the accuracy and F1 score decrease. This validates our hypothesis that poor embeddings impact the quality of learned quantitative judges.

| Dropped features | Offset Bias | | | | Nectar | | | |
|---|---|---|---|---|---|---|---|---|
| | BTL | | BTL2 | | BTL | | BTL2 | |
| [%] | Acc | F1 | Acc | F1 | Acc | F1 | Acc | F1 |
| **Prometheus** 0 | 0.712 | 0.711 | 0.828 | 0.784 | 0.712 | 0.711 | 0.724 | 0.722 |
| 50 | 0.585 | 0.446 | 0.743 | 0.597 | 0.684 | 0.681 | 0.623 | 0.588 |
| 75 | 0.470 | 0.381 | 0.616 | 0.191 | 0.707 | 0.703 | 0.616 | 0.595 |
| 87.5 | 0.575 | 0.419 | 0.574 | 0.006 | 0.679 | 0.667 | 0.569 | 0.466 |
| **Llama** 0 | 0.692 | 0.691 | 0.835 | 0.797 | 0.716 | 0.715 | 0.664 | 0.659 |
| 50 | 0.649 | 0.649 | 0.805 | 0.758 | 0.672 | 0.672 | 0.646 | 0.635 |
| 75 | 0.557 | 0.641 | 0.720 | 0.563 | 0.707 | 0.705 | 0.618 | 0.474 |
| 87.5 | 0.557 | 0.641 | 0.587 | 0.525 | 0.689 | 0.688 | 0.621 | 0.463 |

Table 5: Test accuracy and F1 of BTL and BTL2 judges on preference prediction tasks as a function of embedding quality.

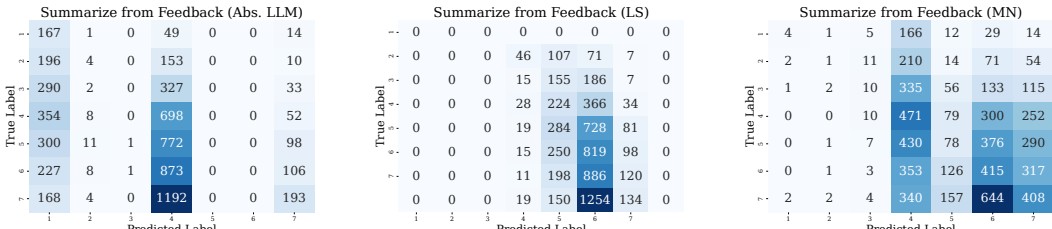

Figure 10: Confusion matrices of base, LS, and MN judges on Summarize from Feedback dataset.

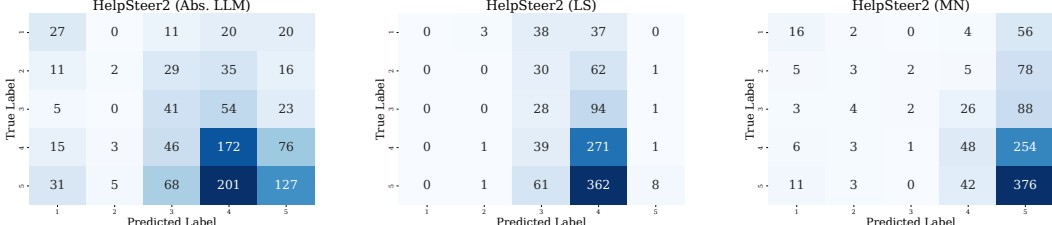

Figure 11: Confusion matrices of base, LS, and MN judges on HelpSteer2 dataset.

### B.4 CONFUSION MATRICES

The confusion matrices of base, LS, and MN judges on Summarize from Feedback and HelpSteer2 datasets are reported in Figures 10 and 11, respectively. These matrices provide a deeper insight on how our judges improve over the base judge in rating prediction tasks. Specifically, we observe that the base judge is poorly calibrated because it is unaware of the score distribution in the domain. For instance, it never predicts Likert scores 5 and 6 in Figure 10, and barely predicts Likert scores 3 and 2 in Figures 10 and 11, respectively. On the other hand, predictions of the LS judge concentrate around the mean score because it minimizes the MSE. The mean score is around 6 in Figure 10 and 4 in Figure 11. The MN judge addresses this limitation by treating Likert scores as separate categories. Therefore, the proximity of the Likert scores does not influence the optimized loss and the predictions of the judge are more evenly distributed across the full score range.

## C COMPUTATION TIMES

Our quantitative judges have both inference and training time components (Section 4). We start with discussing the training time.

| Judge | Summarize from Feedback | HelpSteer2 | Offset Bias | Nectar |
|---|---|---|---|---|
| LS | 0.320 / 0.455 | 0.880 / 0.450 | - | - |
| MN | 0.320 / 0.468 | 0.880 / 0.461 | - | - |
| BTL | - | - | 1.569 / 0.457 | 21.483 / 1.249 |
| BTL2 | - | - | 2.247 / 0.538 | 34.731 / 1.221 |
| SFT | 14.160 | 27.460 | 19.300 | 276.700 |

Table 6: Training times of quantitative judges are separated into embedding computation / training the generalized linear model. The times are reported in GPU minutes.

| Judge | Summarize from Feedback | HelpSteer2 | Offset Bias | Nectar |
|---|---|---|---|---|
| LS | 135.5 / 22.9 / 10.3 | 147.0 / 26.6 / 10.2 | - | - |
| MN | 135.5 / 22.9 / 10.6 | 147.0 / 26.6 / 10.2 | - | - |
| BTL | - | - | 146.1 / 34.5 / 10.2 | 160.2 / 32.5 / 10.2 |
| BTL2 | - | - | 209.1 / 57.2 / 11.3 | 240.7 / 53.8 / 11.4 |

Table 7: Inference times of quantitative judges separated into base judge's run time / embedding computation / GLM inference. The times are reported in GPU seconds per 1000 requests.

We report the total GPU training times of our quantitative judges and SFT in Table 6. The times are measured on an NVIDIA-A100-SXM4-80GB GPU. We observe two major trends. First, the times of our judges are often dominated by that of computing the base judge's embeddings $\phi(e)$. Second, our times could be an order of magnitude smaller than those of SFT. Notably, our training time on Offset Bias dataset is $19.3/(2.247 + 0.538) = 6.93$ times lower and yet we outperform SFT in all metrics in Table 2. This is a direct benefit of the statistical and computational efficiency of simpler models at smaller sample sizes.

We report inference times in Table 7. The most costly part is clearly running the base judge. The cost of embedding its output and using the quantitative judge is about $25\%$ of the total cost. We note that our current implementation computes the embedding separately and not as a byproduct of base judge's inference, which would have a near-zero additional cost. The overhead of quantitative judges in this implementation would be less than $10\%$.

## D  TECHNICAL CONTRIBUTIONS

We extend the BTL2 judge into $K$-way feedback in Appendix D.1 and prove that our judges are not worse than their base judges with a high probability in Appendix D.2.

### D.1  $K$-WAY FEEDBACK IN BTL2

The BTL2 judge needs to be modified at two places: training and inference.

**Training:** Let $\phi(e_1), \ldots, \phi(e_K)$ be the embeddings of base judge's rationales for $K$ responses and $b_1, \ldots, b_K > 0$ be their scores. The responses are ordered such that $b_1 \geq \cdots \geq b_K$. The feature vector of evaluation $k \in [K]$ is $\phi(e_k) \oplus b_k$. With these data, a Placket-Luce model with parameter $\theta \in \mathbb{R}^{d+1}$ is learned to maximize the probability of the observed permutation, which is defined as

$$\prod_{k=1}^{K-1} \frac{\exp[(\phi(e_k) \oplus b_k)^T \theta]}{\sum_{i=k}^{K} \exp[(\phi(e_i) \oplus b_i)^T \theta]} \, .$$

**Inference:** The most probable choice is sampled from a categorical distribution

$$p(k) = \frac{\exp[(\phi(e_k) \oplus b_k)^T \theta]}{\sum_{i=1}^{K} \exp[(\phi(e_i) \oplus b_i)^T \theta]}$$

and returned by the judge.

## D.2   ANALYSIS

We prove that as the sample size $n$ increases, the quantitative judge performs at least as well as its base judge with a high probability. The proof is under the assumption of no regularization.

Let $L(\theta)$ be the expected loss of the quantitative judge with parameter $\theta$ and $L_n(\theta)$ be its empirical loss on a dataset of size $n$ with regularization strength $\gamma = 0$. A standard generalization bound for machine learning models (Murphy, 2012), which also holds for GLMs, says that

$$|L(\theta) - L_n(\theta)| = O\left(\sqrt{C \log(1/\delta)/n}\right)$$

holds for any $\theta$ with probability at least $1 - \delta$, where $C$ is some notion of complexity.

Let $\theta_* = \arg\min_\theta L(\theta)$ and $\hat{\theta} = \arg\min_\theta L_n(\theta)$. From the properties of $\theta_*$ and $\hat{\theta}$, and the triangle inequality, we get that

$$
\begin{aligned}
|L(\hat{\theta}) - L(\theta_*)| &= |L(\hat{\theta}) - L_n(\hat{\theta}) + L_n(\hat{\theta}) - L(\theta_*)| \\
&\leq |L(\hat{\theta}) - L_n(\hat{\theta}) + L_n(\theta_*) - L(\theta_*)| \\
&\leq |L(\hat{\theta}) - L_n(\hat{\theta})| + |L_n(\theta_*) - L(\theta_*)| \\
&= O\left(\sqrt{C \log(1/\delta)/n}\right)
\end{aligned}
$$

holds with probability at least $1 - 2\delta$. Therefore, as $n$ increases, $\hat{\theta}$ performs similarly to $\theta_*$. Now note that $\hat{\theta}$ is the learned quantitative judge parameter. Moreover, the base judge cannot perform better than $\theta_*$ because it can be instantiated using our parameters and $\theta_*$ is the optimal solution. Therefore, as the sample size $n$ increases, the quantitative judge is guaranteed to perform at least as well as its base judge with a high probability.

