# OpenReview forum: "Quantitative LLM Judges"
_ICLR.cc/2026/Conference — Submitted to ICLR 2026_

### Official Review · Reviewer_a3PY · 2025-10-28

**Soundness:** 1
**Presentation:** 2
**Contribution:** 1
**Rating:** 2
**Confidence:** 4

**Summary:**

This paper proposes "*quantitative LLM judges*", where they train a generalized linear model (GLM) on top of the embeddings and score prediction from an LLM-as-a-judge model. They propose several variants for pointwise and pairwise rating, including the least square judge, the multinomial judge, and two variants of Bradley-Terry-Luce (BTL) judges. On four datasets, they show that the quantitative judge is useful and works well when the training data is limited.

**Strengths:**

- This paper proposes a lightweight method to use and improve existing LLM-as-a-judge models
- The framework can be used in both pairwise and pointwise LLM-as-a-Judge tasks
- The method requires fewer training resources, including time and data.

**Weaknesses:**

- The results do not consistently show that the quantitative judge is better than the baselines, including the absolute/relative base (e.g., Table 1 summarize from feedback Llama and Prometheus) or the SFT baselines. While the SFT baseline requires much training time, this might be acceptable since the model only needs to be trained once, and the training overhead is not a problem.
- Lack of comparison to important baselines, including TRACT (which is cited in this paper) and Critique-out-loud reward model (CLoud). The two methods use regression loss to fine-tune for LLM as a judge, where TRACT uses the LM head to calculate the score, and Cloud trains a new regression head to predict the score. TRACT also shows that it can be applied to pairwise and pointwise evaluation datasets, so it seems like a reasonable baseline to be compared with. In line 44, the reviewed paper cites TRACT and argues that LLMs are poorly suited for regressing scores, while TRACT itself is designed to solve the problem. It seems somewhat odd not to include this model for comparison. Moreover, the method described in this paper, the squared judge, looks very similar to Cloud when fixing the LLM and only training the regression head, except that the least square judge in this paper additionally takes the score prediction as the input.
- Lack of comparisons to simpler calibration methods: The proposed method is a method for calibration, where they want to use the original score/judgment prediction and the embedding from the base judge model, and predict a new *calibrated score*, and this calibration model is a small GLM. It is unclear whether it is necessary to learn such a calibration model based on the embedding and score, or whether we can simply use well-known calibration methods, including temperature scaling, to calibrate the score and improve the alignment to human preference.
- Insufficient ablation studies: A critical question is where the performance improvement is from. Is it from the embedding of the judge or from the score prediction? A valid and critical ablation I would like to see is how the performance varies if we only use either the base judge's embedding or the score as the input. How would the performance become? Note that here, using only the score as the input to the GLM becomes the simple baseline mentioned in the previous weakness
- Some unsupported sentences: The last two sentences on page 2 are unsupported by any references.

**Questions:**

- Q1. What does it mean to say "*While LLMs excel at producing qualitative textual evaluations, they often struggle to predict human preferences and numeric scores*". Do you want to say it is good for predicting the CoT (rationale) for the evaluation, but the score may not be calibrated and aligned with the ground truth human score? If this is the case, can this claim be justified? More precisely, is there any past evidence that focuses on the quality of the rationales?
- Q2. What does it mean by "*LLM judges typically output both rationales and numeric scores, thus combining the comprehensiveness of human evaluation with the scalability of automated metrics*"? It seems that there is a causal relation between the two sentences, indicated by the word *"thus"*. However, it is not directly clear how the two sentences are related to each other.
- Some of the papers, including "LLMs instead of Human Judges? A Large Scale Empirical Study across 20 NLP Evaluation Tasks" and "Tract: Regression-aware fine-tuning meets chain-of-thought reasoning for llm-as-a-judge", are from ACL 2025, but they are cited in the arxiv format. It is recommended to cite the papers in their proceedings form. I am only listing the first two entries I found, and I leave checking other papers to the authors.

---

### Official Review · Reviewer_fDpg · 2025-11-01

**Soundness:** 3
**Presentation:** 3
**Contribution:** 3
**Rating:** 6
**Confidence:** 4

**Summary:**

This paper proposed a framework that extends the base judge model by introducing an additional stage after the original rationale and score generation. Specifically, a small model is trained to predict the final score based on the rationale embedding and the initial score produced by the base judge. This extension improves alignment with human evaluations while remaining computationally and data efficient compared to SFT.

**Strengths:**

1. The proposed framework is highly generalizable, as it relies only on rationales and scores and can therefore be applied to any base LLM judge.
2. The paper effectively leverages the original rationales and scores by training a small model to predict the final score, improving alignment with human evaluations in a computationally and data-efficient manner.
3. The paper tests various types of small models, such as the least-squares (LS) judge, multinomial (MN) judge, Bradley–Terry–Luce (BTL) judge, and two-headed BTL (BTL2) judge, to demonstrate the effectiveness of their framework.

**Weaknesses:**

1. It is helpful to include experiments that use a larger and more complex model for the quantitative judges, such as a two-layer MLP.
2. The paper currently lacks ablation studies and analyses investigating the impact of the rationale embedding and the initial score. In particular, it is necessary to include experiments that use only rationale embeddings and those that use only the score.

**Questions:**

1. How is the rationale embedding $\phi(e)$ specifically derived from the base judge?
2. For embedding quality, it is helpful to use the Qwen3 Embedding [1] or other advanced embedding models [2] to provide a more convincing demonstration.

References

[1] https://huggingface.co/collections/Qwen/qwen3-embedding

[2] https://huggingface.co/spaces/mteb/leaderboard

---

### Official Review · Reviewer_6cZR · 2025-11-01

**Soundness:** 2
**Presentation:** 2
**Contribution:** 2
**Rating:** 4
**Confidence:** 3

**Summary:**

This paper proposes a framework for calibrating LLM-as-a-judge on human preference data. In particular, the method relies on the use of a lightweight model (GLMs and variants) to align an LLM’s numeric scores with human preferences. In empirical experiments, the quantitative LLM judges outperform base and fine-tuned LLM judges in several instances and require less computation than fine-tuning.

**Strengths:**

-	The calibration framework presented is simple, post hoc, and easy to implement.

-	It is (pleasantly) surprising that a lightweight model such as ordinary least squares is sufficient to calibrate LLM-judges to human evaluations. In a similar vein, it is nice that this method is post hoc (efficient to optimize a GLM, no fine-tuning required). The proposed calibration framework is simple, elegant, and easy to implement.

-	The paper is very well motivated, addressing the important and timely concern that LLM judges often fail to align with human preferences and can exhibit structural biases (e.g., favoring responses from models with similar architectures or training data). The authors do a nice job situating their work within this broader landscape of prior studies.

**Weaknesses:**

-The proposed quantitative judges are trained and evaluated on datasets labeled by the same human preferences used to assess their alignment. This setup is circular: models are optimized to reproduce human scores and then evaluated on their ability to approximate those same scores. To ensure that the method is not simply overfitting to the human labels, a stronger experiment might evaluate calibration on held-out datasets (can be in a similar domain).

-Dependence on Human Evaluations: The approach relies on the availability of human evaluation data for calibration, which can be challenging to obtain at scale, especially in settings where LLM-as-a-judge is intended to reduce human effort.

-The study evaluates only two base models. It would be helpful to discuss or test whether the proposed calibration framework generalizes to models of different sizes or architectures.

-Please provide a justification for the design choice of concatenating the judge’s score with its rationale. Why is this a natural or theoretically grounded way of combining the two signals? The concatenation of rationales with judge scores is an intuitive step, but there is no analysis of how much the textual component versus the numerical score contributes to predictive power. This makes it unclear whether the system is actually “reasoning better” or simply leveraging linguistic correlations in rationales.

-In Section 4.1, the concept of “population bias” is introduced without clear explanation. Please provide a clear moviation for each new term, when it is introduced in the paper.

Minor Stylistic /Presentation Revisions:

-The phrasing “Comparison to base judges and fine-tuning them” reads colloquially; consider “Comparison to base and fine-tuned judges.”

-Please ensure that font usage is consistent near the end of Section 5.2.

**Questions:**

Please see weaknesses.

---

### Official Review · Reviewer_t8ti · 2025-11-03

**Soundness:** 2
**Presentation:** 3
**Contribution:** 2
**Rating:** 4
**Confidence:** 4

**Summary:**

The paper proposes an evaluation method based on the LLMs-as-Judges framework. The central idea is to introduce an additional trainable module that maps the reasoning and judgment outputs of the base LLM to a final evaluation score. Depending on the evaluation paradigm, this module can take different forms. The paper addresses both pointwise and pairwise evaluation settings and experiments across a broad range of datasets.

**Strengths:**

- The paper is clearly written and well structured, with comprehensive coverage of both model architectures and datasets.

- The proposed approach is conceptually straightforward and easy to follow.

**Weaknesses:**

- The novelty of the proposed method appears limited. Prior research has already explored the use of trainable adapters or probing modules to enhance evaluation accuracy.

- The experimental comparison is relatively weak. The method is compared with only one SFT baseline. To convincingly demonstrate superior performance, the paper should include a broader set of baselines—both training-free calibration methods (e.g., G-Eval) and approaches that incorporate additional training modules, such as linear probing methods [1].

[1] Improving Preference Extraction In LLMs By Identifying Latent Knowledge Through Classifying Probes

**Questions:**

Is it possible to extend the framework to unsupervised evaluation?

---

### Meta-Review · Area_Chair_VDhd · 2025-12-28

**Summary:**

This paper presents methods for postprocessing the predictions of LLM judges and turning them into more reliable quantitative scores.  All of these methods combine features of the explanation (phi(e)) with the base model score into a vector that allows the learned judge to at least mimic the original judge and ideally improve on it.  The paper compares these methods to basic SFT on explanations and scores across a number of settings.

Strengths:

- The paper clearly lays out a number of potential models for LLM judges

- Results demonstrate that fairly simple models can work surprisingly well

Weaknesses:

- The wins over the SFT baseline are not strong or uniform across settings

- Lack of comparison with stronger baselines (see reviewer concerns)

- It's not clear whether the forms of these judges matter. It's nice that they can give the same outputs as the initial judge; this seems like a useful form of regularization. But the benefits of this model structure are not demonstrated, and it's not clear if there is a reason to prefer these judge models over others despite their apparent simplicity.

**Reviewer Concerns:**

Performance of the method doesn't consistently win over SFT (a3PY): "The results do not consistently show that the quantitative judge is better than the baselines"

Lack of comparison with baselines (a3PY, t8ti): There are two issues here. The first is lack of comparison to simple methods such as temperature scaling. The second is lack of comparison to TRACT and CLoud, or other known methods for balancing losses for predicting explanations paired with scores, such as having a separate head or reweighting losses.

Unclear motivation for the design of the judges (6cZR).

No rebuttal was offered so none of these are addressed.

**Reviewer Scores:**

No rebuttal given.

---

### Decision · Program_Chairs · 2026-01-26

Reject